# Towards general network architecture design criteria for negative gas adsorption transitions in ultraporous frameworks

Simon Krause [1], Jack D. Evans [1,2], Volodymyr Bon [1], Irena Senkovska[1], Paul Iacomi[3], Felicitas Kolbe[1], Sebastian Ehrling[1], Erik Troschke[1], Jürgen Getzschmann[1], Daniel M. Többens[4], Alexandra Franz[4], Dirk Wallacher [4], Pascal G. Yot[5], Guillaume Maurin[5], Eike Brunner[1], Philip L. Llewellyn[3], François-Xavier Coudert [2] & Stefan Kaskel [1]

Switchable metal-organic frameworks (MOFs) have been proposed for various energy-related storage and separation applications, but the mechanistic understanding of adsorption-induced switching transitions is still at an early stage. Here we report critical design criteria for negative gas adsorption (NGA), a counterintuitive feature of pressure amplifying materials, hitherto uniquely observed in a highly porous framework compound (DUT-49). These criteria are derived by analysing the physical effects of micromechanics, pore size, interpenetration, adsorption enthalpies, and the pore filling mechanism using advanced in situ X-ray and neutron diffraction, NMR spectroscopy, and calorimetric techniques parallelised to adsorption for a series of six isoreticular networks. Aided by computational modelling, we identify DUT-50 as a new pressure amplifying material featuring distinct NGA transitions upon methane and argon adsorption. In situ neutron diffraction analysis of the methane ($CD_4$) adsorption sites at 111 K supported by grand canonical Monte Carlo simulations reveals a sudden population of the largest mesopore to be the critical filling step initiating structural contraction and NGA. In contrast, interpenetration leads to framework stiffening and specific pore volume reduction, both factors effectively suppressing NGA transitions.

[1] Faculty of Chemistry and Food Chemistry, TU Dresden, Bergstrasse 66, 01062 Dresden, Germany. [2] Chimie ParisTech, PSL University, CNRS, Institut de Recherche de Chimie, Paris, 75005 Paris, France. [3] Aix-Marseille Univ., CNRS, MADIREL (UMR 7246), 13013 Marseille, France. [4] Helmholtz-Zentrum Berlin für Materialien und Energie, Hahn-Meitner-Platz 1, 14109 Berlin, Germany. [5] Institut Charles Gerhardt Montpellier UMR 5253 Univ. Montpellier CNRS UM ENSCM, Université de Montpellier, Place Eugène Bataillon, 34095 Montpellier cedex 05, France. Correspondence and requests for materials should be addressed to S.K. (email: stefan.kaskel@tu-dresden.de)

P orous solids featuring adaptable pore sizes as a stimulated response to characteristic molecules in the gas phase, i.e. changes of gas pressure or activity, led to the discovery of unique phenomena such as gating[1], breathing[2], and more recently negative gas adsorption (NGA)[3]. Metal-organic frameworks (MOFs) reach record values of porosity[4–6] (specific surface area, pore volume), often at the expense of reduced mechanical stability[7]. Hence, energetics at the solid–fluid interface reach an order of magnitude comparable with chemical bond energetics leading to stimuli responsive structural transformations and severe deformations of the framework constituents. A stark contrast in adsorption behaviour compared with rigid adsorbents[8–11] has promoted new applications in gas separation[12,13], storage[14], proton conduction[15], and sensing[16]. However, the mechanistic understanding and rational governing responsivity in such soft porous solids is still at an early stage[17–20]. The experimental discovery of flexibility in MOFs has often been dictated by serendipity while rational synthesis strategies have rarely been applied[21]. Rationalisation of adsorption-induced structural transformations has been achieved by analysing their thermodynamics via density functional theory (DFT)[20] and molecular dynamic (MD) simulations[22], demonstrating the importance of micromechanics in responsive frameworks. But modular construction of MOFs provides an ideal platform to achieve a mechanistic understanding of adsorption-induced structural deformations. Isoreticular pore expansion is a powerful modular concept[23] leading to prototypical series of solids with individually adjustable porosity and flexibility[24–27]. The versatility of this approach provides clear structure–property relationships and mechanistic insights by independently probing the impact of surface functionality, mechanical stiffness and porosity.

NGA is a novel counterintuitive phenomenon discovered in a highly porous framework[3]. DUT-49, a MOF constructed by connecting 9H-carbazole-3,6-dicarboxylate-based metal-organic polyhedra (MOP) with 4,4′-substituted 1,1′-biphenyl, was originally synthesised for methane storage applications[28]. However, upon adsorption of gases such as methane (111 K) or n-butane (298 K)[3] a colossal structural contraction is induced, accompanied by an external gas pressure increase. NGA materials show self-propelled gas pressure amplification. Upon structural contraction, DUT-49 expels previously adsorbed gas from the framework leading to a stepwise desorption in the adsorption branch of the isotherm corresponding to a pressure amplification in the closed measuring cell[3]. Such NGA transitions require the system to traverse through metastable states. Experimental and computational analysis of NGA in DUT-49 revealed enhanced solid–fluid interactions for the contracted pore structure (cp) to be the driving force for the contraction of the initially stable open pore state (op)[29]. The structural transitions during NGA for xenon (200 K) were confirmed by in situ $^{129}$Xe nuclear magnetic resonance (NMR) spectroscopy[30], while for nitrogen (77 K), a subtle influence of crystallite size indicates NGA to be a highly cooperative phenomenon, as structural transitions are suppressed in small crystallites (<1 μm)[31]. Reducing the pore size and ligand length in an isoreticular network (DUT-48) increases the framework rigidity and decreases the energetic driving force for the contraction, suppressing the adsorption-induced structural transition[32], while the external pressure, inducing the compaction is increased from 35 MPa (DUT-49) to 65 MPa (DUT-48) using mercury as the pressure transducing medium.

Until today, DUT-49 has been the only known framework material showing NGA transitions. In the following we deduce rational design criteria for NGA, prerequisites from the perspective of the network structure, using a combined theoretical and experimental approach. Systematic linker elongation in a series of six isoreticular frameworks shows the crucial importance

of pore size, framework softness, and interpenetration to permit adsorption-induced structural contraction leading to NGA transitions. The computational prediction is confirmed by experimental analysis of the structural transitions using advanced in situ diffraction, calorimetric, and spectroscopic experimentation. Ultimately, we deduce DUT-50 as the second NGA material among more than 20 000 MOFs known today. This finding indicates NGA to be a general phenomenon observable for a wider class of highly porous materials satisfying specific structural design rules.

## Results

**In silico network micromechanics**. The assembly of modular porous networks and prediction of their properties for separation and storage of gaseous energy carriers is nowadays effectively aided by computational modelling[33–36]. Consequently, before investing major synthetic efforts, we decided to model an isoreticular series of DUT-48-derived MOFs with increasing ligand length. Based on the experimental crystal structures of DUT-48[32] and DUT-49[28], the number of 1,4-substituted phenylene units in the ligand backbone was increased step-wise from one in $L_1$ (DUT-48, Fig. 1) and two in $L_3$ (DUT-49) to three in $L_4$ (DUT-50) and four units in $L_5$ (DUT-151). In addition, we decided to include a 2,6-substituted naphthalene unit in $L_2$ (DUT-46) to achieve an intermediate length between $L_1$ and $L_3$. As a result, a series of five MOFs was obtained all sharing the same topology and trimodal pore structure with lattice constants increasing from 39.91 to 58.55 Å and pore volumes ranging from 1.8 to 4.6 cm$^3$ g$^{-1}$ (DUT-48 to DUT-151, Fig. 1, Supplementary Data 1).

The compression of empty DUT-49op requires significant energy and is dominated by the buckling deformation of the ligand. For first insights into micromechanics we analysed the buckling behaviour of isolated ligands $L_1$–$L_5$ using DFT methods by applying compressive strain decreasing the N–N distance from the local minimum to a compressive strain of 0.05 (Fig. 2).

For all ligands, the initial elastic deformation regime is followed by severe inelastic ligand deflection at higher compressive strain. As expected, the shorter ligands withstand higher stress ($L_1$: 3.40 nN, $L_2$: 1.99 nN) before buckling occurs. While buckling of $L_4$ is observed at 1.06 nN in the range of $L_3$ (1.04 nN), $L_5$ demonstrates buckling at even lower stress (0.68 nN), indicating that these ligands and consequently the MOFs should demonstrate softer behaviour similar to DUT-49. Because the ligands comprise the same phenylene-based backbone with varying length, all ligands display equivalent elastic moduli and the mechanical behaviour can be exclusively attributed to the variation in length. To ascertain whether the behaviour of a single ligand is transferable to the behaviour of the corresponding crystalline network, we analysed the compressibility and free energy profile of a whole unit cell by computationally less expensive classical methods using force fields which were found to describe the behaviour of DUT-49 well[29,37]. The free energy profiles of the series of MOFs were all found to exhibit a bistable nature, a crucial requirement for NGA, with activation energies in a comparable range[38]. However, transition pressures and energies for the structural contraction were found to decrease with increasing ligand length (Supplementary Table 31). Consequently, ligand elongation in this system should favour adsorption-induced structural contraction and potentially NGA, assuming that the adsorption-induced stress in these systems is of comparable magnitude.

**Synthesis of ligands and MOFs**. A retrosynthetic approach based on previously described synthesis of carbazole-derived carboxylates[39] was developed to obtain the proposed ligands and MOF materials. Ligands $L_1$–$L_5$ could be obtained in six-step syntheses

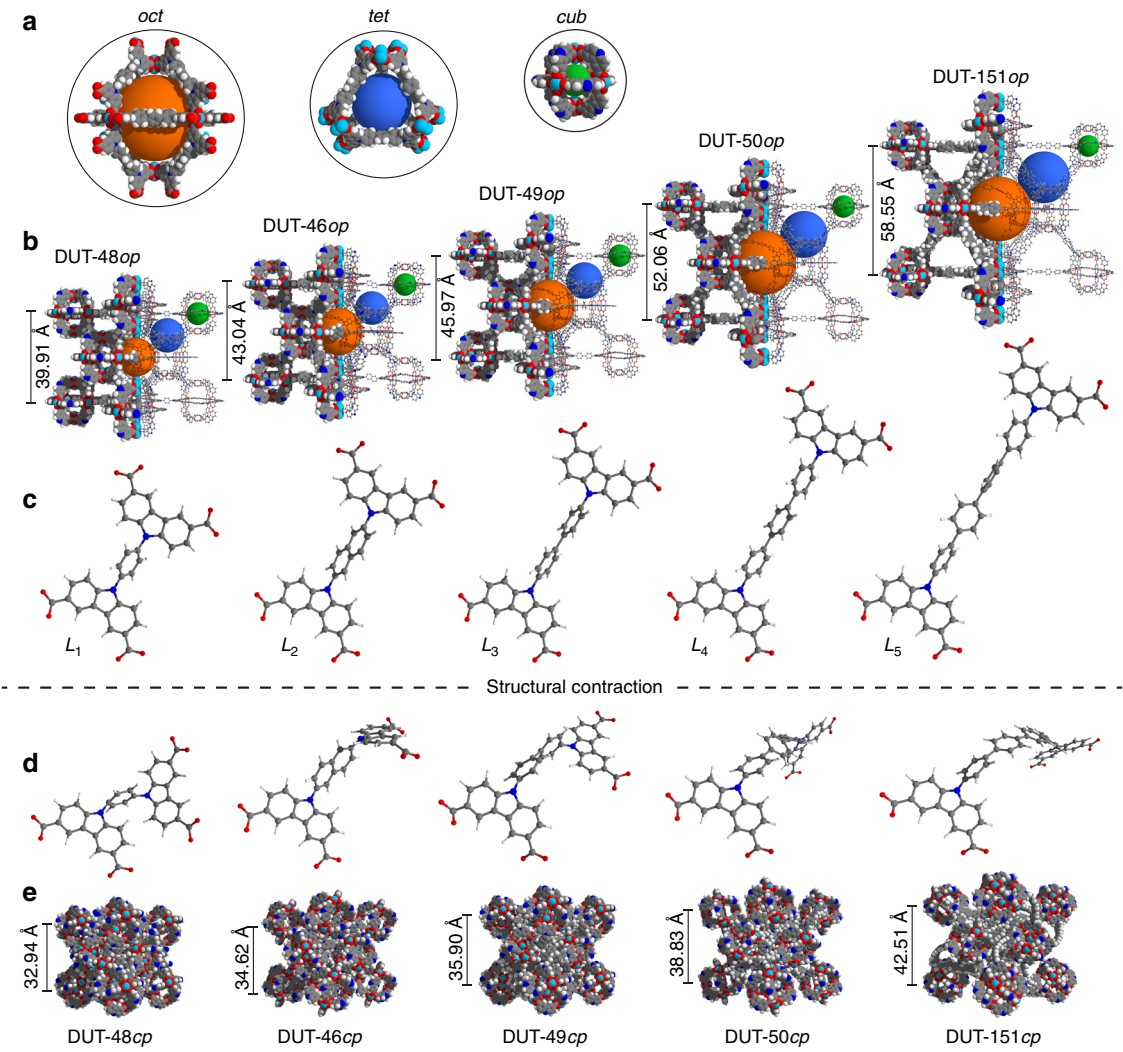

**Fig. 1** Crystal structures and ligands of isoreticular DUT-49 derivatives. **a** Trimodal pore system: octahedral (oct, orange), tetrahedral (tet, blue), cuboctahedral (cub, green). **b** Crystal structures of open pore (*op*) of DUT-48, -46, -49, -50, and non-interpenetrated DUT-151 from left to right including respective lattice parameter of the cubic unit cells with $Fm\bar{3}m$ symmetry. Conformation of ligands $L_1$–$L_5$ before (**c**) and after (**d**) structural contraction. **e** Crystal structures of contracted pore *(cp)* structures of DUT-48, 46, 49, 50, and non-interpenetrated DUT-151 from left to right including respective lattice parameter of the cubic unit cells with $Pa\bar{3}$ symmetry. Lattice constants for the structures were obtained by relaxation using force fields which were found to describe the behaviour of DUT-49 well[29, 37]. Colour code: hydrogen (white), carbon (grey), nitrogen (blue), oxygen (red), and copper (turquois)

with total yields of over 60% (for detailed information see Supplementary Note 1). The crystalline network syntheses were elaborated to yield phase pure crystalline powders of the targeted MOFs as confirmed by powder X-ray diffraction (PXRD) (Supplementary Note 2, Supplementary Fig. 5), thermogravimetric analysis (Supplementary Fig. 6), elemental analysis (Supplementary Table 5), and diffuse reflectance infrared Fourier transform (DRIFT) spectroscopy (Supplementary Fig. 59). Desolvation of the MOF powders was performed using a protocol previously applied on DUT-49[3]. Particle size distributions were determined via scanning electron microscopy (SEM) and the mean crystal size was found to exceed 2.5 µm (Supplementary Table 6). The latter is important, as we recently demonstrated that NGA critically depends on crystallite size for DUT-49[31].

The structures of DUT-46, -50, and -151 were determined using synchrotron-based single crystal X-ray diffraction experiments (Supplementary Note 3). They crystallise in the cubic space group $Fm\bar{3}m$, except DUT-151 which crystallises in the monoclinic space group $C2/m$ containing two interwoven **fcu** nets (Supplementary Fig. 31). The latter finding is interesting for two

reasons: First, it demonstrates limitations of ligand elongation to obtain a non-interpenetrated structure in the isoreticular series of DUT-49 at given synthesis conditions. Second, the impact of interpenetration on structural contraction and NGA can be analysed. Consequently, the computational analysis of DUT-151 was extended by the experimentally obtained interpenetrated crystal structure DUT-151*int*. The free energy profile no longer exhibits a bistable nature in the energy range of the non-interpenetrated structures which can be expected for a much denser structure that lacks the void required for the ligand buckling upon contraction (Fig. 2g, h).

**Experimental analysis of structural responsivity**. The response of the guest-free MOF powders towards external mechanical pressure was experimentally analysed using mercury as a pressure transducing medium, revealing an upper stress limit for compaction and the corresponding volume changes[32] (Supplementary Note 6, Supplementary Fig. 17). With increasing linker length the transition pressures corresponding to the compression of the

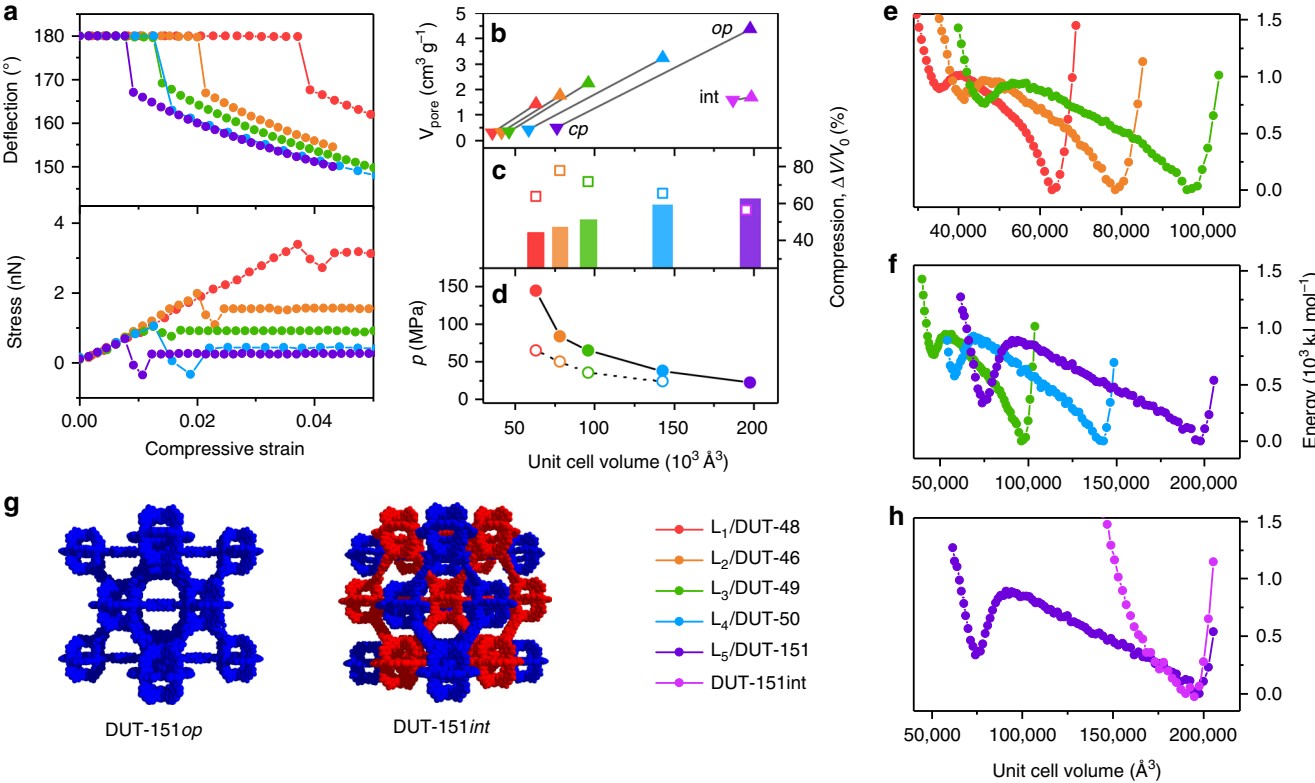

**Fig. 2** Mechanical properties of ligands and MOFs. **a** DFT analysis of ligands $L_1$ (red), $L_2$ (orange), $L_3$ (green), $L_4$ (blue), and $L_5$ (purple). **b** Evolution of pore volume upon contraction in the open (triangles up) and closed (triangles down) states. **c** Compression by hydrostatic pressure (open squares correspond to Hg intrusion experiments). **d** Transition pressures under hydrostatic compression (open circles correspond to Hg intrusion experiments). **e**, **f** Free energy profiles of non-interpenetrated MOFs per unit cell. **g** Structures of non- and double-interpenetrated DUT-151. **h** Free energy profiles of DUT-151 and DUT-151*int*. Colour code: DUT-48 (red), DUT-46 (orange), DUT-49 (green), DUT-50 (blue), DUT-151 (purple) and DUT-151*int* (pink). Closed symbols represent simulation, open symbols experimental data

frameworks decrease from 65 (DUT-48) to 24 MPa (DUT-50). The experimental pressures follow the trend of the simulated values but are found to be in general lower, however within the range of accuracy of these methods (Fig. 2d). The volume change upon compression was found to be larger than simulated values with higher deviation for DUT-48 and DUT-47, indicating a complete compaction of the networks for both, while a good agreement for an *op–cp* transition is observed for DUT-50 (Fig. 2c). However, reliable phase analysis after compression is hampered by mercury contamination. Previously, we have demonstrated the differences between experimental and simulated compression for DUT-48 and DUT-49 originating from amorphization of the samples[32]. Hg intrusion appears to produce greater compression possibly causing bond breakage not captured by the classical simulations used. The gradual intrusion curve observed for DUT-151*int* (Supplementary Fig. 17) with an initial transition pressure of 29 MPa and a total volumetric compression of only 1.53 cm³ g⁻¹ indicates a transition mechanism different compared with the non-interpenetrated structures, for example caused by inter-framework dynamics.

In order to elucidate structural design criteria provoking NGA and pressure amplification in the networks, we analysed the new network series with respect to adsorption-induced structural contractions by exposing them to gases such as *n*-butane (273, 298, 303 K, Supplementary Fig. 13, Supplementary Fig. 15, Supplementary Table 9) and methane (111 K, Supplementary Fig. 14)[3]. Nitrogen (77 K) and argon (87 K) adsorption were used to characterise textural properties such as surface area and pore volume (Supplementary Fig. 12). Methane adsorption at 111 K and argon adsorption at 87 K finally allowed us to uncover DUT-

50 as a new material showing pronounced NGA transitions and hysteresis similar to DUT-49, while none of the other networks exhibits NGA. DUT-46 showed no hysteresis and the isotherm is similar to DUT-48. Isotherms of DUT-151*int* exhibit small hysteresis but absence of NGA due to a higher rigidity and lack of bistability (Fig. 2h).

In situ calorimetry during the adsorption and desorption of *n*-butane at 303 K allowed us to measure the differential enthalpy of adsorption $\Delta_{ads}H$ and thus experimentally analyse the adsorption energetics during NGA at near ambient conditions (Supplementary Note 5, Fig. 3). For the entire series of non-interpenetrated structures, the enthalpy profile during adsorption exhibits a maximum at $p/p_0 \approx 0.01$. With increasing pressure, a reduction and minimum of $|\Delta_{ads}H|$ is observed shifting to increasing $p/p_0$ with increasing ligands length (DUT-48: $p/p_0 = 0.06$, DUT-50: $p/p_0 = 0.2$). A subsequent sudden increase in $|\Delta_{ads}H|$ correlates with a steep slope in uptake in the isotherm due to enhanced fluid–fluid interactions at complete pore filling. At lower $p/p_0$, the enthalpy profiles are identical for the investigated series reflecting similar adsorption mechanisms at lower loadings. However, with increasing pore size, the confinement of the guest is expected to decrease,[40] as is well reflected by the observed reduction in $|\Delta_{ads}H|$ with increasing ligand length and corresponding pore volume of the MOF (Fig. 3f). In DUT-46, a non-hysteretic profile is observed for both isotherm and enthalpy branches (Fig. 3a). In DUT-49 and DUT-50, an increase in enthalpy is observed upon desorption after NGA alongside a hysteresis in the isotherm. Thus, the analysis of the desorption branch provides information about the adsorption enthalpy of the *cp*-phase. The difference of enthalpy between the *op* and *cp* structure was previously found to

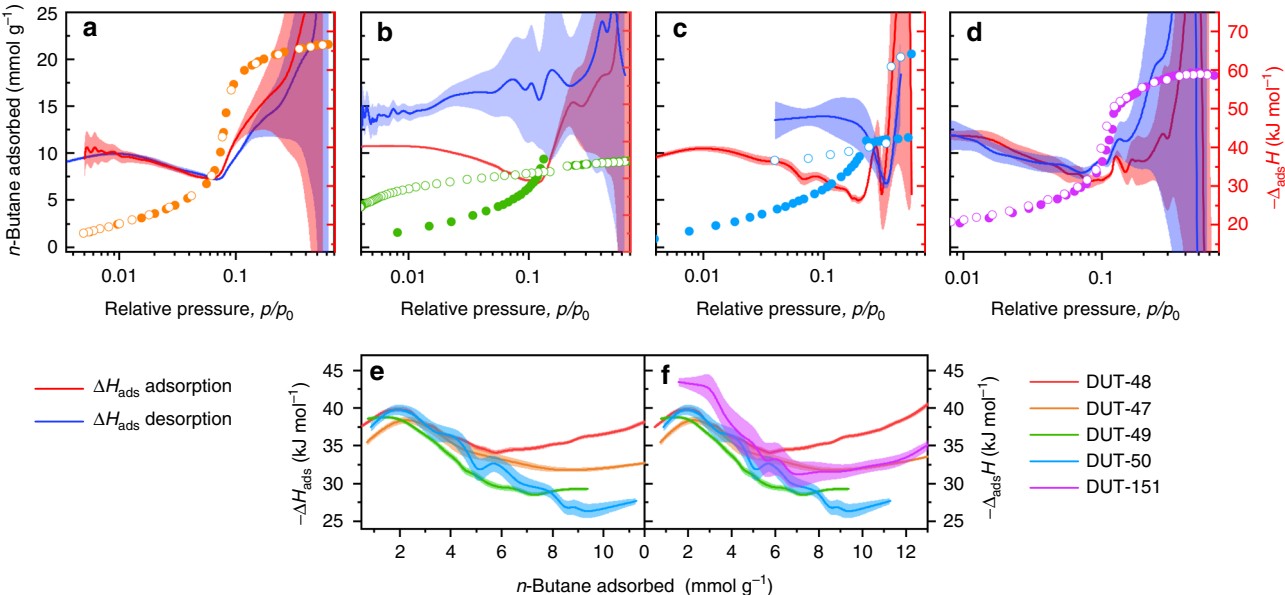

**Fig. 3** Analysis of *n*-butane adsorption energetics. in situ calorimetric analysis in parallel to adsorption/desorption of *n*-butane at 303 K for **a** DUT-46, **b** DUT-49, **c** DUT-50, and **d** DUT-151*int*. Adsorption and desorption in closed and open symbols, respectively, adsorption enthalpies red, enthalpy of desorption blue, error regimes indicated as coloured areas. **e**, **f** Adsorption enthalpies plotted against adsorbed amount of *n*-butane for the series non-interpenetrated MOFs (**e**) and in comparison with DUT-151*int* (**f**)

be the driving force of the structural contraction[3] and is now for the first time accessible experimentally in the pressure range of NGA. The difference in adsorption enthalpy between the *op* and *cp* structures ($25.4 \pm 2$ kJ mol$^{-1}$) is multiplied by the number of *n*-butane molecules ($n_{ads}$) adsorbed per unit cell in the *op*-phase to give $\Delta\Delta_{ads}H_{total}$, which approximates the gain in enthalpy per unit cell upon structural contraction (Supplementary Note 11). Interestingly, upon adsorption of *n*-butane in DUT-49, $\Delta\Delta_{ads}H_{total} = -4470 \pm 550$ kJ mol$_{uc}^{-1}$ (mol$_{uc}$ refers 1 mol unit cells of the respective DUT), which exceeds the transition energy calculated for guest-free DUT-49 by over 2500 kJ mol$_{uc}^{-1}$ (Fig. 2f). In comparison, DUT-151*int* shows a higher adsorption enthalpy throughout the whole pressure range, reflecting the smaller pore size and enhanced solid–fluid interactions. A small hysteresis at a relative pressure of 0.06–0.11 indicates possible network displacements of the two interpenetrating nets (Supplementary Fig. 16).

To analyse the structural transitions further we performed in situ PXRD experiments in parallel to adsorption of *n*-butane at 273 K. DUT-50 shows significant peak broadening and loss of crystallinity upon adsorption of *n*-butane at 20 kPa and 273 K (Fig. 4c, Supplementary Fig. 19, Supplementary Note 7).

At elevated pressures the peaks of the pristine *op*-phase reappear, reflecting the reversible reopening of the structure indicating a rare crystalline-disordered-crystalline transition. Upon desorption, the *op*-phase undergoes contraction without indications for reopening at lower pressure. The fact that DUT-50 completely transforms into the *op*-phase at higher pressures suggests that the framework connectivity is preserved and is an evidence for a cooperative transformation within a single crystal. Reduction in crystallinity upon DUT-50 contraction is more pronounced in comparison with DUT-49 (Fig. 4c) indicating a higher degree of flexibility of the ligand and loss of long-range order of the building units within the highly porous network providing a higher degree of freedom for displacements of the building blocks. For complementary understanding of the local structural transformations we analysed the series of MOFs by in situ DRIFT (Supplementary Note 9, Supplementary Fig. 60,

Supplementary Fig. 66) and solid-state $^{13}$C cross-polarisation (CP) MAS NMR spectroscopy (Supplementary Note 10, Supplementary Fig. 67, Supplementary Fig. 70) in parallel to adsorption of *n*-butane at 298 K. Both methods are ideal to probe local chemical environments independent of long-range order and crystallinity. Indeed, only a small degree of peak broadening in the spectra of both methods is observed upon contraction indicating a uniform ligand conformation in the *cp*-phase in both DUT-49 and 50. Shifts especially of peaks assigned to C–H groups of the bridging units indicate uniform changes in the local environment associated with the buckling illustrated in Fig. 1. In addition, two signals in the $^{13}$C CP MAS NMR spectra at lower chemical shifts are observed upon contraction in DUT-50 (Fig. 4c). They can be assigned to the aliphatic carbons of *n*-butane indicating a partial immobilisation of *n*-butane within the contracted pores. In contrast, spectra of *n*-butane loaded DUT-48 and DUT-151*int* which do not undergo large scale structural contraction do not exhibit these peaks, demonstrating the rapid diffusion of the *n*-butane within the open pores of these systems. Comparison of the DRIFT spectra for *op* and *cp* structures in DUT-49 and DUT-50 (Supplementary Fig. 62, 63) demonstrate red shifts of several peaks that can be attributed to elongation of the C–N and C–C bonds in the bridging unit of the ligands. Vibrations of the carbazole-core remain mostly unchanged indicating that the structure of the MOPs remains unchanged upon structural contraction. Although the spectroscopic analysis supports that the structural contraction in DUT-50 is of similar nature as in DUT-49, the dynamics of DUT-50*cp* at 273 K obstruct an in depth crystallographic structural analysis.

As this high degree of mobility within the strained ligands may be reduced at lower temperature, we decided to analyse the adsorption behaviour with in situ PXRD for methane at 111 K (Supplementary Note 7, Supplementary Fig. 20) as an NGA inducing medium. Again, a strong reduction of crystallinity and peak broadening is observed after NGA, however less severe compared with the patterns recorded in the presence of *n*-butane at 273 K. The unit cell of DUT-50*cp*, appearing after NGA was refined by Le Bail method in monoclinic symmetry ($P2_1/c$ space

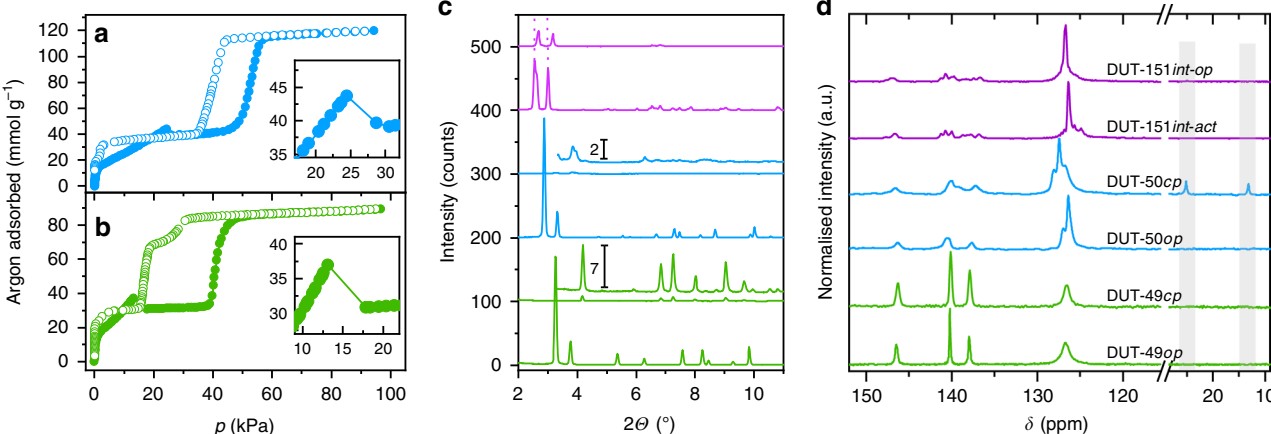

**Fig. 4** Argon adsorption, in situ X-ray diffraction and NMR spectroscopy. **a**, **b** Argon adsorption (filled circles) and desorption (open circles) isotherm at 87 K of DUT-50 (**a**) and DUT-49 (**b**), inset represents magnified region of NGA. **c** X-ray diffraction patterns of evacuated (bottom) and *n*-butane loaded (top) samples at 273 K including magnification (offset 100 counts). **d** $^{13}$C CP MAS NMR spectra of argon-filled (bottom) and *n*-butane loaded (top) samples at 298 K, grey areas indicate peaks corresponding to *n*-butane. Colour code: DUT-49 (green), DUT-50 (blue), DUT-151*int* (purple)

group). The unit cell volume of 60611 Å$^3$ represents a 59% volume reduction compared with DUT-50*op*. The latter is in good agreement with DUT-50*cp* predicted in silico ($V = 58185$ Å$^3$, Fig. 2e and Supplementary Table 31). Moreover, during desorption there is evidence for intermediate pore phases, similar to those observed in DUT-49[3]. In summary, DUT-50 undergoes NGA transitions analogous to those of DUT-49, as demonstrated using various in situ methods confirming the predicted structural models.

For DUT-151*int* only minor structural flexibility is observed, as predicted by in silico modelling. The PXRD patterns of DUT-151*int* after supercritical activation (DUT-151*int-act*) show a shift of peaks in the range of 2–3.5° to lower $2\Theta$ values and appearance of new reflections in comparison with the as made material are observed. This change can be assigned to a change in symmetry and unit cell volume upon solvent removal from monoclinic $C2/m$ (in solvated DUT-151*int*) to triclinic $P\bar{1}$ in DUT-151*int-act*. Adsorption of *n*-butane at 273 K at intermediate pressures (8–10 kPa) causes reformation of the initial monoclinic net ($C2/m$), however the structure is contracted by 14% (DUT-151*int-cp*, 87419 Å$^3$) compared with DUT-151*int* (101103 Å$^3$). With increasing pressure, the unit cell volume increases again reaching 101740 Å$^3$ (DUT-151*int-op*). The reversed path is observed during desorption (For structural details see Supplementary Fig. 31). Overall, the ligand deformations are less severe and the difference in pore volume of DUT-151*int-op* vs. DUT-151*int-cp* is only 0.32 cm$^3$ g$^{-1}$, less than 20% compared with the colossal contractions in DUT-49 and DUT-50. DUT-151*int* lacks the pronounced bistability required for NGA materials as predicted in silico (Fig. 2h) and confirmed by in situ PXRD (Supplementary Table 14 and Supplementary Fig. 27).

**in situ analysis of the adsorbate structure**. In regard to the adsorption mechanism of NGA two open questions remain: (i) what is the role of characteristic adsorbate structures forming metastable states and their role in pressure amplification phenomena in mesoporous networks, and (ii) from which pore of the frameworks is the gas released upon NGA. To locate methane molecules within the pores of DUT-49 and derivatives at different pressures/loadings, we decided to combine computational grand canonical Monte Carlo (GCMC) analysis of methane with experimental in situ neutron powder diffraction (NPD) in parallel to the adsorption of deuterated methane (CD$_4$) at 111 K (Supplementary Note 8). NPD has previously been used to investigate

primary adsorption sites in MOFs via Rietveld refinement[41–44], however the analysis of higher loadings in mesoporous networks at elevated temperatures such as 111 K, crucial for mechanistic understanding of NGA, remains unexplored. Cho et al. recently analysed the pore filling of individual pores by combined gas adsorption and in situ X-ray diffraction[45].

Methane adsorption isotherms and corresponding adsorption enthalpy profiles at 111 K for *op* and *cp* phases were simulated by GCMC methods (Supplementary Note 12, Supplementary Fig. 71). A good agreement between the simulated and the experimental isotherms (Supplementary Fig. 14) indicates the validity of the simulations which allow us to further analyse the adsorption energetics. The enthalpy profile of DUT-151*int* is different in nature compared with the non-interpenetrated MOFs due to the different pore structure and will thus be excluded from the following discussion. Simulated enthalpy profiles of all non-interpenetrated *op* phases are almost identical up to 1.1 kPa where a maximum at $-\Delta_{ads}H_{op} = 14.3$ kJ mol$^{-1}$ is reached. At higher pressures a decrease in $-\Delta_{ads}H_{op}$ is observed reaching a minimum at 11.2 kJ mol$^{-1}$ at 3.7 kPa for DUT-48, $-\Delta_{ads}H = 10.2$ kJ mol$^{-1}$ at 10.3 kPa for DUT-49, and 9.6 kJ mol$^{-1}$ at 23 kPa for DUT-50. In this pressure region, the adsorption enthalpies ($-\Delta_{ads}H_{cp}$) of the corresponding *cp* phases are found to decrease with increasing ligand length from 17.8 kJ mol$^{-1}$ in DUT-48 to 16.7 kJ mol$^{-1}$ in DUT-50. With increasing ligand length, the adsorbed amount of methane per unit cell at the intersection of the *op* and *cp* isotherm increases from 318 CH$_4$ per UC (DUT-48), 388 CH$_4$ per UC (DUT-46), 463 CH$_4$ per UC (DUT-49) to 596 CH$_4$ per UC (DUT-50). Consequently, the values of $\Delta\Delta_{ads}H_{total}$ at this point are found to favour structural contraction as they become progressively more exothermic (Supplementary Note 11, Supplementary Table 32): $-1177$ kJ mol$_{uc}^{-1}$ (DUT-48), $-2289$ kJ mol$_{uc}^{-1}$ (DUT-46), $-3491$ kJ mol$_{uc}^{-1}$ (DUT-49), to $-4458$ kJ mol$_{uc}^{-1}$ (DUT-50).

To derive the impact of different pore sizes on NGA experimentally we analysed at least six loadings of CD$_4$ in a wide pressure range from 0.1 to 20 kPa for DUT-48, -49, and -50 at 111 K and refined the CD$_4$ positions by Rietveld analysis taking single crystal structural data of the guest-free frameworks and complementary computational analysis into account for analysing a multitude of loadings in regions of the isotherms experimentally not accessible due to structural transitions (Fig. 5).

The analysis shows that for all three frameworks methane is first adsorbed in the cuboctahedral micropores (cub) identical in

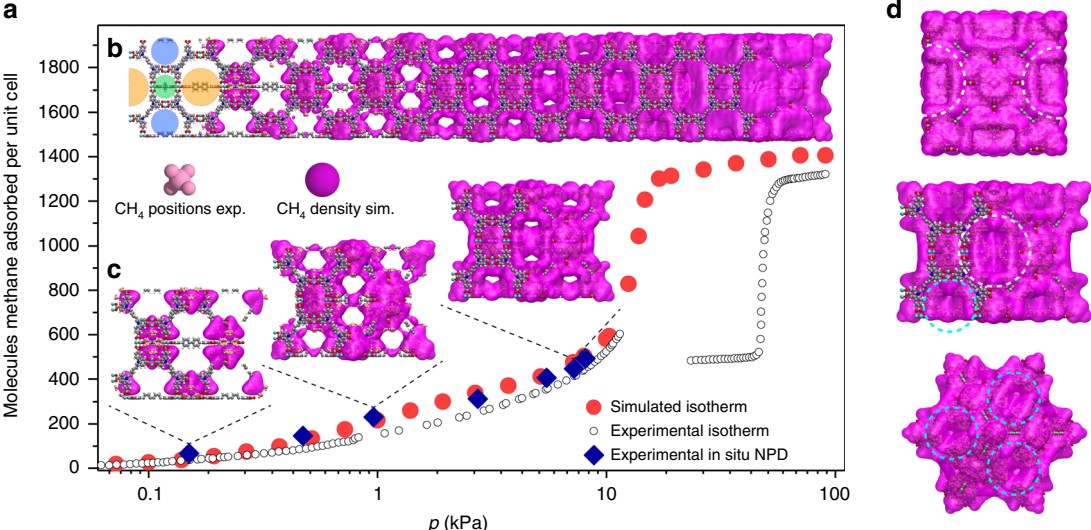

**Fig. 5** Methane distribution in DUT-49. **a** Simulated (red circles) and experimental (open symbols) methane adsorption isotherm of DUT-49 at 111 K including points at which NPD patterns were recorded (blue diamonds). **b** Structure of DUT-49 with increasing population of pores with methane upon pressure increase, including on the left trimodal pore system: octahedral (oct, orange), tetrahedral (tet, blue), cuboctahedral (cub, green). Experimentally refined methane positions in light pink, simulated methane density in pink. **c** Visualisation of DUT-49 unit cell at selected loadings which correspond to experimentally investigated pressure. **d** Illustration of methane-filled pores of DUT-49 unit cell in the region of NGA viewed along (100) direction (top), (110) direction (middle), and (111) direction (bottom). For detailed mechanism see Supplementary Movie 1

size for each of the investigated MOFs. Primary adsorption sites are the open metal sites inside the cub pore and its pore windows. This is in line with the analysis of other MOFs with similar structure[44]. At higher loadings (>300 $CD_4$ per unit cell), visualisation and analysis of adsorption sites in the 3D unit cells becomes challenging due to the large amount of $CD_4$ molecules. Illustration of the methane positions by methane distribution maps (Supplementary Movie 1) provides a qualitative picture of the pore filling but make extraction of a quantitative pore filling mechanism difficult. Thus, crystal structures with refined $CD_4$ positions from Rietveld refinements (Supplementary Data 2) as well as methane distributions obtained by GCMC analysis were investigated by radial distribution analysis. This statistical method, in which each methane molecule is assigned to be present in one pore by analysing its distance to the respective pore centre, allows us to dissect isotherms of the *op* phases in separate isotherms for each pore. In the case of DUT-49 and its derivatives a trimodal pore structure consisting of cuboctahedral (cub, 1.0 nm), tetrahedral (tet, 1.4–2.1 nm), and octahedral (oct, 1.9–3.1 nm) voids was extracted from the pore size distributions (Supplementary Fig. 73) and used as a model for the analysis of the pore filling. In addition, the extracted single-pore isotherms can be correlated to the adsorption enthalpy profiles of the *op* phases of DUT-48, -49 and -50 to derive the adsorption sites responsible for the non-monotonic profile (Fig. 6).

in situ NPD and simulations for DUT-48, -49 and -50 revealed the cub voids to be filled first with methane reaching saturation at around 1 kPa. Thus, the low-pressure adsorption enthalpies are dominated by solid–fluid interactions in the cub pore and the gradual increase correlates with additional fluid–fluid interactions upon pore filling. In the range of 0.7−2 kPa tet and oct voids are starting to be filled with methane molecules adsorbing on the linker backbone framing these pores. In this region the adsorption enthalpy increases more strongly due to additional solid–fluid interactions reaching a maximum around 1.1 kPa where the cub pore reaches saturation. At higher pressures the decrease in adsorption enthalpy can be correlated with reduced solid–fluid interactions due to multilayer adsorption on the surface of the tet and oct voids.

Upon NGA ($p_{NGA} = 11.7$ kPa in DUT-49, $p_{NGA} = 19.5$ kPa in DUT-50) neither tet nor oct voids have reached saturation. Interestingly, a steep slope in the isotherm and enthalpy profile which indicates collective pore filling typical for capillary condensation in mesoporous materials[46,47] is found for both voids in both materials at pressures slightly higher than $p_{NGA}$. This collective filling is more pronounced for the oct compared with the tet voids due to larger respective pore sizes and shifted towards higher pressures with increasing pore size. Adsorption-induced stress and deformation in mesoporous materials with cylindrical pores typically reaches a maximum in the region of collective pore filling referred to as capillary condensation[48]. By analysing the evolution of pore size distribution upon contraction for DUT-48−151 in the *op* and corresponding *cp* phases, the number of molecules present in the cub and tet voids of the *op* phases and *cp* phases before and after NGA are found to be almost identical. Thus, we can conclude that methane released upon NGA primarily originates from fluid adsorbed in the oct pore, which undergoes the largest reduction in pore volume upon structural contraction. In addition, the performed analysis indicates that the filling of the cub pore does not affect NGA and thus a material with bimodal or possibly monomodal pore system and adaptive pore sizes in the range found for DUT-49−151 should demonstrate a similar adsorption mechanism. However, a secondary role of cub pores assisting delayed capillary condensation by hampering molecular guest redistribution cannot be completely ruled out.

## Discussion

In this work, we evince NGA as a counterintuitive phenomenon not limited to the single network DUT-49. Systematic linker expansion in an isoreticular series reveals an expanded version, DUT-50, with even higher pore volume and larger pore size to show NGA transitions closely related to those found in DUT-49. The series analysis allows us to derive structural prerequisites from the framework point of view to promote NGA. Two minima in the free energy profile of the empty host corresponding to structures with pronounced porosity differences separated by an

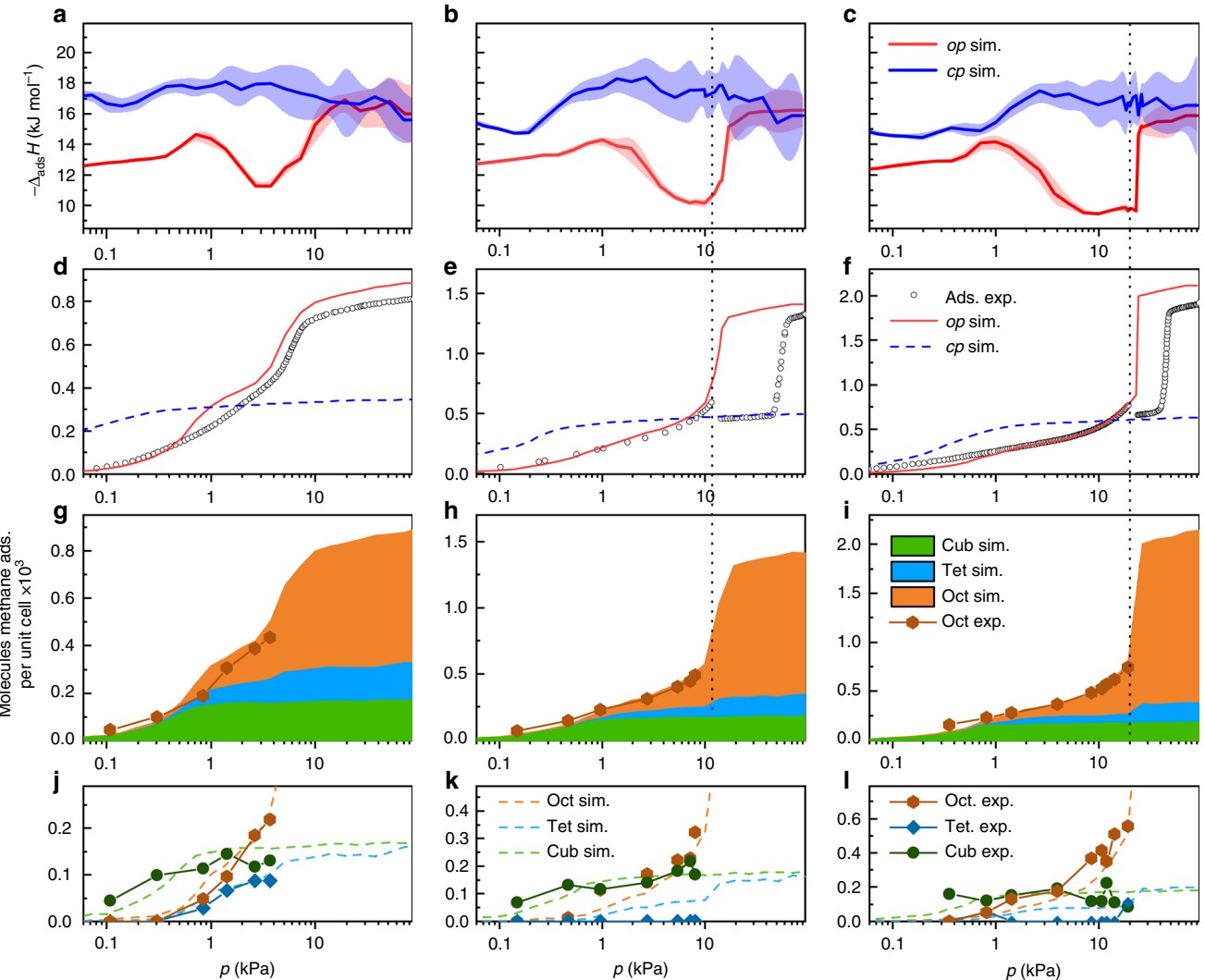

**Fig. 6** Analysis of adsorption mechanism. Columns represent DUT-48 (left), DUT-49 (middle), and DUT-50 (right). **a–c** Adsorption enthalpy profiles for *op* (red) and *cp* (blue). **d–f** Methane adsorption isotherms at 111 K experimental (circles), simulated for *op* (red) and *cp* (blue dashed line). **g–i** Stacked simulated isotherms of methane at 111 K for the cuboctahedral (green), tetrahedral (blue), and octahedral (orange) pores and experimentally obtained isotherms (orange hexagons). **j–l** Comparison of simulated (dashed lines) and experimental (symbols and solid lines) for the cuboctahedral (dark green), tetrahedral (dark blue), and octahedral (dark orange) pores. Vertical dashed black lines indicate the region of NGA for DUT-49 and -50

activation barrier are a necessary condition for pressure amplifying materials and NGA. Using a combined computational and experimental analysis of the mechanical stability, we conclude linker elongation as a factor reducing the stress required to stimulate structural contraction. Thus, only DUT-49, and -50 show adsorption-induced contraction while the adsorption stress is not high enough to contract the more robust networks DUT-48 and -46. The interpenetrated structure of DUT-151*int* shows detectable adsorption-induced deformation but lacks the free pore volume change and bistability for colossal contraction.

Analysis of the adsorption mechanism of methane in DUT-48, -49 and -50 by in situ NPD in combination with GCMC methods correlates the filling of the pores and adsorption sites to the adsorption enthalpy profiles. As neither the tet nor the oct voids have reached saturation at $p_{NGA}$, their filling by fluid condensation and contraction is the main reason for NGA transitions. This is consistent with DUT-50 exhibiting a higher $p_{NGA}$ than DUT-49, owing to the large pore sizes responsible for delayed mesopore filling. Hence, we suggest a pore size above 2 nm is a prerequisite for NGA. Generic computational slit pore models for adsorption-induced deformation in dynamic mesoporous systems support

this hypothesis[38]. Methane released during NGA was found to mainly originate from the largest octahedral cavity which implicates their pore volume to correlate with $\Delta n_{NGA}$. However, $\Delta n_{NGA}$ in DUT-50 is found to be lower than in DUT-49 indicating a somewhat decreased activation barrier defining the metastability regime of DUT-50*op*. This difference in activation barrier for the solid transformation can be rationalised considering the lower mechanical stiffness of DUT-50. Hence, an increase of specific pore volume and mechanical stiffness are both equally important to maximise $\Delta n_{NGA}$ and pressure amplification of mesoporous frameworks in future.

## Methods

**Instruments**. NMR spectra were acquired on a BRUKER Avance III 500 spectrometer (500.13/600.16 MHz and 125.77/150.91 MHz for $^1$H and $^{13}$C respectively) and/or on a VARIAN Mercury (300 MHz, 282 MHz and 75.5 MHz for $^1$H and $^{13}$C, respectively). All $^1$H and $^{13}$C NMR spectra are reported in parts per million (ppm) downfield of TMS and were measured relative to the residual signals of the solvents at 7.26 ppm (CHCl$_3$) or 2.54 ppm (DMSO). Data for $^1$H NMR spectra are described as following: chemical shift ($\delta$ (ppm)), multiplicity (s, singlet; d, doublet; t, triplet; q, quartet; m, multiplet; br, broad signal), coupling constant $J$ (Hz), integration corresponding to amount of C or CH. Data for $^{13}$C NMR spectra are

described in terms of chemical shift (δ (ppm)) and functionality was derived from DEPT spectra.

Matrix-assisted laser desorption/ionisation (MALDI) time of flight (TOF) mass spectrometry analysis was performed on a BRUKER Autoflex Speed MALDI TOF MS using dithranol as matrix. Atmospheric-pressure solid analysis probe mass spectrometry was performed on an ADVION expression LCMS with an APCI ion source.

Elemental analysis was carried out on a VARIO MICRO-cube Elemental Analyser by Elementar Analysatorsysteme GmbH in CHNS modus. The composition was determined as the average of three individual measurements on three individually prepared samples.

DRIFT spectroscopy was performed on a BRUKER VERTEX 70 with a SPECAC Golden Gate DRIFT setup. Prior to the measurement 2 mg of sample was mixed with 10–15 mg dry KBr in a mortar and pressed in the DRIFT-cell. Assignments of peaks in wavenumber ν (cm$^{-1}$) were categorised by strong (s), medium (m) and weak (w).

Thermal analysis was carried out in synthetic dry air using a NETZSCH STA 409 thermal analyser at a heating rate of 5 K min$^{-1}$. Air sensitive MOF samples were prepared in an Ar-filled glove box and inserted in the instrument with little exposure to ambient conditions.

PXRD patterns were collected in transmission geometry with a STOE STADI P diffractometer operated at 40 kV and 30 mA with monochromatic Cu-Kα$_1$ ($\lambda = 0.15405$ nm) radiation, a scan speed of 30–15 s/step and a detector step size of $2\Theta = 0.1–2°$. The samples were placed between non-diffracting adhesive tape or in a glass capillary. "As made" samples were analysed while suspended in DMF. Desolvated samples were prepared under inert atmosphere. Theoretical PXRD patterns were calculated on the basis of crystal structures using Mercury 3.9 software package.

SEM images of DUT-49 were taken with secondary electrons in a HITACHI SU8020 microscope using 1.0 kV acceleration voltage and 10.8 mm working distance. The powdered samples were prepared on a sticky carbon sample holder. To avoid degradation upon exposure to air, the samples were prepared under argon atmosphere. For each sample a series of images was recorded at different magnifications and for each sample three different spots on the sample holder were investigated. The crystal size refers to the edge length of the cubic crystals as they are the easiest to measure. The analysis of the SEM images was performed with ImageJ Software package[49]. Values for mean crystal size, as well as relative standard deviation was obtained by using the ImageJ Analyse-Distribution function.

**Synthesis of the ligands**. All chemicals (Supplementary Table 1) and gases used in the synthesis and analysis in this study were of high purity. Attempts to synthesise L$_3$-derived ligands by using the synthesis conditions previously used to synthesise L$_3$ involving lithiation and subsequent carboxylation proved difficult for synthesising the proposed series of ligands mainly due to low solubility of the bromides and low yields (Supplementary Note 1). Thus, we decided to apply a different synthetic pathway (Supplementary Fig. 1). The same approach was previously used in the synthesis of L$_1$ in DUT-48[32] and PCN-81[44,50] (Supplementary Fig. 2). We decided to use a protocol established by Eddaoudi et al.[39]. developed to synthesise 9H-Carbazole-3,6-dicarboxylic acid in large quantities based on the commercially available, inexpensive 9H-carbazole. By esterification with n-butanol, dibutyl 9H-carbazole-3,6-dicarboxylate could be obtained on the scale of 45 g product and overall yield of 71% over four steps. The following reaction sequences were used to synthesise L$_2$–L$_5$ (Supplementary Fig. 4). Based on whether the coupling reaction was performed with an iodide or bromide, reaction conditions were adjusted.

*Ullmann coupling with iodides:* This procedure is based on syntheses previously used for carbazole based ligands[51]. A Schlenk flask was charged with indicated amounts of dibutyl 9H-carbazole-3,6-dicarboxylate, the corresponding iodide, potassium carbonate, copper (I) iodide, and L-proline under inert atmosphere. To the mixture indicated amounts of degassed DMSO or DMF were given and Ar was bubbled through the suspension for 30 min. The reaction mixture was stirred at 90–120 °C for 24 h to 10 days and the reaction was cooled down to room temperature. The suspension was quenched with diluted (<0.02 M) hydrochloric acid and extracted with chloroform. The organic phases were collected, dried over MgSO$_4$, and the solvent removed in vacuum. The crude product was purified by flash column chromatography using indicated mixtures of chloroform, DCM, iso-hexane and ethyl acetate. Corresponding amounts of the chemicals added and used for the synthesis and purification are provided for each coupling product in the Supplementary Note 1.

*Ullmann coupling with bromides:* This procedure is based on syntheses previously used for carbazole based ligands[50]. A Schlenk flask was charged with indicated amounts of dibutyl 9H-carbazole-3,6-dicarboxylate, the corresponding bromide, potassium carbonate, copper (I) iodide, and N,N'-dimethylethylenediamine under inert atmosphere. To the mixture indicated amounts of degassed anhydrous 1,4-dioxane were given and Ar was bubbled through the suspension for 30 min. The reaction mixture was stirred at 80–110 °C for 24 h to 12 days and the reaction was cooled down to room temperature. The solvent was removed in vacuum and the obtained powder was dissolved in chloroform and extracted with diluted (<0.02 M) hydrochloric acid. The organic phases were collected, dried over MgSO$_4$, and the solvent removed in vacuum. The crude product was purified by flash column chromatography using indicated mixtures of chloroform, DCM, iso-hexane and ethyl

acetate. Corresponding amounts of the chemicals added and used for the synthesis and purification, reaction times and temperatures are provided for each coupling product in the Supplementary Note 1.

*General procedure for ester hydrolysis:* To hydrolyse the ester groups the corresponding coupling products were dissolved in indicated volumes of THF, methanol, and water at 85 °C. To the solution potassium hydroxide was added and the mixture was stirred at 85 °C for 12 h to 5 days. In case a precipitate formed from the previous clear solution (most likely the potassium salt of the hydrolysed ester which is insoluble in THF) small amounts of water were added until a clear solution formed again. After the indicated reaction time THF and methanol were removed in vacuum, the resulting solution was filtered, and neutralised with 2 M hydrochloric acid. The precipitate was filtered off and dried in vacuum at room temperature. Corresponding amounts of the chemicals added and reaction times are provided for each hydrolysis product in the Supplementary Note 1.

**Synthesis of microcrystalline MOF powders**. In a previous study we have demonstrated that large crystals of DUT-49 show enhanced adsorption capacity and pronounced NGA[31]. Thus, the reaction conditions for the synthesis of DUT-48, -46, -50, -151 were chosen to produce crystals of average size above 2 μm, based on the reaction of DUT-49 by using acetic acid as modulator in a solvothermal reaction of ligands L$_1$–L$_5$ at 80 °C (Supplementary Note 2). Because NMP was found to partially reduce Cu$^{2+}$ if the reactions were carried out over 72 h or longer, we chose to use DMF instead. Due to the lower solubility especially of ligand L$_1$ and L$_2$ in DMF, larger amounts of DMF in comparison with NMP were used. The reaction conditions including the corresponding mean crystal size of the activated MOF powders are summarised in Supplementary Table 2.

**Synthesis of MOF single crystals**. To obtain crystals with sizes large enough for single crystal X-ray diffraction (>90 μm) we adapted the reaction conditions by diluting the reaction mixture with DMF and increasing the amount of acetic acid acting as modulator. The reaction conditions are summarised in Supplementary Table 3.

**Single crystal X-ray diffraction**. Blue cubic single crystals of DUT-46, DUT-50 and DUT-151 with dimensions ranging from 30 μm to 80 μm were prepared in a borosilicate glass capillary ($d = 0.3$ mm) with small amount of the mother liquor (Supplementary Note 3). The capillaries were sealed with wax from both sides in order to avoid the contact with ambient atmosphere. The datasets were collected at BESSY MX BL14.3 beamline of Helmholtz-Zentrum Berlin für Materialien und Energie[52]. Monochromatic X-ray radiation with a wavelength of $\lambda = 0.089499$ nm ($E = 13.85$ keV) was used in experiments. All datasets were collected at room temperature. After short test scans, the crystal symmetry and scan range were determined in each particular case using iMosflm programme[53,54]. The $\varphi$-scans with oscillation range of 0.5° were used for data collection. In the case of cubic structures of DUT-46 and DUT-50, 100 images were collected to reach the maximal completeness. For DUT-151, crystallising in C-centered monoclinic lattice, 240 images were required. Further the datasets were processed automatically using XDSAPP 2.0 software[55]. Crystal structures were solved by direct methods and refined by full matrix least-squares on $F^2$ using SHELX-2016/4 programme package[56,57]. All non-hydrogen atoms were refined in anisotropic approximation. Hydrogen atoms were refined in geometrically calculated positions using "riding model" with $U_{iso}(H) = 1.2 U_{iso}(C)$. Since the symmetry of the naphthalene core in DUT-46 is incompatible with the symmetry of its position in the unit cell, the molecular fragment was refined disordered over four symmetrically dependent positions. In the case of DUT-50, the disorder of both symmetrically independent phenyl rings was treated by splitting over two equally occupied positions. The lower symmetry of DUT-151int, with four symmetrically independent paddle wheels in the asymmetric unit prompted us to use the AFIX 66, SIMU and DELU instructions in order to constrain the geometry and anisotropic displacement parameters of all phenyl rings in the structure. The large pores, high crystal symmetry and high measurement temperature did not allow refining the disordered solvent molecules within the pores of the frameworks, therefore, SQUEEZE routine in PLATON was used to generate the reflection intensities with subtracted solvent contribution[58]. Crystallographic data are summarised in Supplementary Table 4. CCDC-1889257, 1889255 and 1889256 contain the supplementary crystallographic data for DUT-46, DUT-50 and DUT-151int, correspondingly. These data can be obtained free of charge from the Cambridge Crystallographic Data Centre via www.ccdc.cam.ac.uk/data_request/cif.

**Desolvation of microcrystalline MOF powders**. After the solvothermal reaction the blue precipitates were separated from the reaction solution by centrifugation and washed six times with fresh DMF over a period of two days at room temperature. DMF was exchanged with anhydr. acetone by washing 10 times over a period of at least four days. All samples were subjected to an activation procedure involving supercritical CO$_2$, as previously described for DUT-49:[3] In acetone suspended samples were placed on filter frits in a Jumbo Critical Point Dryer 13200 J AB (SPI Supplies) which was subsequently filled with liquid CO$_2$ (99.995% purity) at 15 °C and 50 bar. To ensure a complete substitution of acetone by CO$_2$, the liquid in the autoclave was exchanged with fresh CO$_2$ 18 times over a period of

5 days. Temperature and pressure were then raised beyond the supercritical point of $CO_2$ (35 °C and 100 bar) and kept until the temperature and pressure were constant at least for 1 h. Supercritical $CO_2$ was slowly released over 3 h and the dry powder was transferred and stored in an argon-filled glove box. To ensure complete removal of the solvent from the pores (especially from the open metal sites of the Cu-paddle-wheels) additional thermal activation at 120 °C in a Schlenk-tube under dynamic vacuum of $10^{-4}$ kPa for at least 24 h was performed.

**in situ calorimetry**. For microcalorimetry, all isotherms and enthalpies were measured experimentally using a Tian-Calvet type microcalorimeter coupled with a home-made manometric gas dosing system (Supplementary Note 5)[59]. This apparatus allows the simultaneous measurement of the adsorption isotherm and the corresponding differential enthalpies. Gas is introduced into the system using a step-by-step method and each dose is allowed to stabilise in a reference volume before being brought into contact with the adsorbent located in the microcalorimeter. The introduction of the adsorbate to the sample is accompanied by an exothermic thermal signal, measured by the thermopiles of the microcalorimeter. The peak in the calorimetric signal is integrated over time to give the total energy released during this adsorption step. Around 0.05 g of sample is used in each experiment. For each injection of gas, equilibrium was assumed to have been reached after 130 min. This was confirmed by the return of the calorimetric signal to its baseline (<5 μW). The gases used for the adsorption experiment were obtained from Air Liquide and were of minimum N47 quality (99.997% purity).

**Mercury intrusion**. Mercury intrusion is a method frequently applied to investigate the mechanical properties and structural contraction of frameworks[60–65]. To begin with, the solids were activated at 115 °C for 8 h in secondary vacuum (Supplementary Note 6). The obtained powder was then loaded into a powder penetrometer of 3.1126 $cm^3$ volume with a stem volume of 0.412 $cm^3$ using a glove box (Jacomex P-BOX) under argon atmosphere $H_2O$ < 5 ppm. The mercury intrusion experiments were carried out using a Micromeritics AutoPore IV 9500 allowing a range of pressure applied from 0.003 to 300 MPa. Prior to the experiment the powder was outgassed in vacuum at ~6.5 Pa for 15 min. The collected volume of mercury intruded was corrected by a blank measurement recorded under the same conditions (temperature and pressure) using the same penetrometer to obtain the absolute contracted volume as a function of the pressure reported in Supplementary Fig. 17.

**in situ DRIFT spectroscopy**. DRIFT spectroscopy was performed on a ʙʀᴜᴋᴇʀ VERTEX 70 with a SPECAC Golden Gate DRIFT setup (Supplementary Note 9). Samples were analysed in a HARRICK Praying Mantis reaction chamber sealed by a removable dome with IR-transparent ZnSe windows. The same setup was used to perform in situ DRIFT analysis by applying a constant flow of a mixture of inert carrier gas ($N_2$) and adsorptive (n-butane or vaporised $CCl_4$). For the analysis of n-butane adsorption, the gas composition was controlled via two mass flow controllers. The composition during in situ adsorption of vaporised $CCl_4$ was estimated by two valves controlling the flux of inert and $CCl_4$ saturated inert gas.

**Solid-state MAS NMR studies**. Solid-state NMR measurements were carried out on a Bruker Ascend 800 spectrometer equipped with a 3.2 mm HX probe (Supplementary Note 10). The $^1H$–$^{13}C$ cross-polarisation experiments were performed at a resonance frequency of 800.2 MHz for protons and 201.2 MHz for carbon. The contact time was 4 ms and the relaxation delay was 3 s. A sample rotation of 15 kHz was chosen. The $^{13}C$ chemical shift was referenced using adamantane. The rotor was filled with the activated material (op-phase) under argon atmosphere. After NMR analysis of argon-filled DUT-49op, the rotor was opened under n-butane atmosphere and equilibrated for 2 min to obtain the cp-phase of the MOF. Formation of the cp-phase under these conditions was confirmed in a separate experiment by in situ PXRD. After sealing the rotor, the measurements of the butane filled cp-phase were carried out. Subsequently, the rotor was re-opened allowing the sample to have extended contact to ambient conditions and moisture which is known to initiate decomposition of the material. For these samples no reflections in the PXRD patterns could be observed.

**in situ powder X-ray diffraction**. in situ PXRD studies and parallelised gas adsorption were measured at KMC-2 beamline[66] of the BESSY II synchrotron, operated by Helmholtz-Zentrum Berlin für Materialien und Energie (Supplementary Note 7). Self-designed automated instrumentation, based on the volumetric adsorption instrument and closed-cycle helium cryostat, equipped with adsorption chamber with beryllium domes was used in all experiments[67]. PXRD patterns were measured at constant wavelength $\lambda = 0.15406$ nm ($E = 8048$ eV) in transmission geometry. Because of the bulky cryostat, the sample holder cannot rotate during experiments, however an average crystallite size in the range of 2–15 μm and using an area 2D detector (Vantec 2000, Bruker) allowed to record diffraction images with reasonable particle statistics. Each 2D image was measured with 31 s exposure. For each experiment 10–12 mg of sample were used. In order to block the reflections coming from the crystalline Be-dome, a tungsten aperture with 5 mm opening was mounted on the detector cone. The obtained diffraction images were integrated using DATASQUEEZE 2.2.9[68] with further processing in

FITYK 0.9[69] software. PXRD patterns, measured for DUT-50 during adsorption and desorption of methane at 111 K (Supplementary Fig. 20) and n-butane at 273 K (Supplementary Fig. 19) were obtained in the automatic mode in parallel to the isotherm measurement. The dataset of DUT-151int tracking the adsorption and desorption of n-butane at 273 K was collected in the automated modus as well (Supplementary Fig. 22). For all automated measurements the physisorption isotherms were measured using equilibrium settings for pressure change of 0.1% within 300 s. The dataset during n-butane adsorption at 303 K (Supplementary Fig. 18) as well as the analysis of DUT-151int at 273 K (Supplementary Fig. 22) was measured in the manual mode. Each pressure was set manually and PXRD patterns were measured after the pressure in the cell was stabilised for at least 300 s.

**in situ neutron powder diffraction**. in situ NPD experiments were conducted at neutron powder diffractometer E9-FIREPOD at the BER II neutron source of the Helmholtz-Zentrum Berlin[70] using incoming neutrons with wavelength $\lambda = 2.8172$ (2) Å (Supplementary Note 8). A cylindrically-shaped aluminium sample holder with inner diameter of 1 cm and length of 12 cm was filled and sealed under argon atmosphere with 130, 115, and 95 mg of DUT-48, −49, −50, respectively. The sample holder was connected to a gas handling system via a stainless steel capillary that allowed controlled dosing of methane in the pressure range of 0.001–200 kPa with a home-made apparatus. The sample was cooled by a closed-cycle helium cryostat system and the temperature was monitored directly at the sample cell with a Lakeshore temperature controller with accuracy of 0.1 K. Prior to the NPD measurement the samples were outgassed in dynamic vacuum (<0.1 Pa) at 298 K for at least 1 h. Afterwards the samples were cooled to 111 K and diffractograms of the guest-free evacuated MOFs were recorded. Based on the $CH_4$ isotherms at 111 K, pressures were selected to achieve different loadings of $CD_4$. Selected loadings/pressures are displayed in Supplementary Fig. 32. Further information about the refinement of the obtained diffraction data is provided in the supplementary information.

**Simulation of structural models of MOFs**. The structural models were simulated using Material Studio 5.0 software package[71]. Initial structural models for DUT-46, DUT-50 and DUT-151 are based on the experimental single crystal structures of DUT-49[28] and DUT-48[32]. Because these experimental crystal structures contain disordered phenylene backbones, disorder-free models were obtained by reducing the crystal symmetry from $Fm\overline{3}m$ to $Pa\overline{3}$ and subsequent removal of the disordered positions. For simulation of DUT-46 the phenylene group in DUT-48 was substituted by a 2,6-naphthylene unit. DUT-50 was modelled by adding an additional phenylene unit to the ligand in DUT-49. For the non-interpenetrated structure of DUT-151, which could not be obtained experimentally, an additional phenyl ring was inserted in the ligand backbone of DUT-50. The geometry of the structure and lattice parameters including the disorder-free structures of DUT-48 and DUT-49 were optimised using UFF force field, integrated into Material Studio 5.0. Subsequently, the simulated structural models were reduced in symmetry to P1, optimised using the MOF-FF force field[37] as described below (Molecular Dynamics Simulations). Final lattice parameters were taken from (N, V, T) simulations of the op and cp minima (see Fig. 1 main text) and the energy and structures were minimised iteratively adjusting the coordinates and cell parameters using the lammps minimise and box/relax commands with default convergence criteria (Supplementary Table 30, Supplementary Data 1).

**Simulation of pore size distribution**. Pore size distributions (PSD) were simulated for these structures using the Zeo++ code[72] on the basis of in silico determined crystal structures for the op and cp phases for DUT-48–151 (Supplementary Note 4, Supplementary Fig. 73). In addition, PSD for various refined phases of DUT-151int were performed on the experimentally refined crystal structures (Supplementary Fig. 74). Pore volume and pore sizes are summarised in Supplementary Table 33 and surface area and pore volume in comparison with experimental results are summarised in Supplementary Table 7.

**Density functional theory simulations**. Ligands L$_1$–L$_5$ were simulated as the corresponding acids using DFT employed by the CRYSTAL14 software[73]. Localised TZVP basis sets[74] and the hybrid exchange-correlation functional PBE0[75] were used. Dispersion corrections were included using the Grimme "D2" approach. Ligand structures were strained by a decrease in the N–N length from the local minimum to a compressive strain of 0.06, in 40 steps. For each step, the structure was optimised with the N–N length fixed. Subsequently, a stress-strain curve relative to this axial compressive deformation of the ligand was generated; stress is defined by the gradient of the energy, and strain is the relative decrease in N–N length.

**Molecular dynamics simulations**. Molecular dynamics simulations to produce equations of states for DUT-46, DUT-48, DUT-49, DUT-50, DUT-151int and DUT-151 used a modified MOF-FF force field[37] adapted to lammps[76,77] to describe the bonds, angles, dihedrals and improper dihedrals present in the frameworks. Further details can be found in the supporting information. Representative input files for molecular simulations are available online in our data repository at https://github.com/fxcoudert/citable-data. Structural models of the

*cp*-phase were obtained from the MD simulation at the local minimum of reduced unit cell volume. The final structural models are based on geometrical optimisation using the above described force fields in the space group *P*1. Structure parameters are summarised in Supplementary Table 31.

**Grand canonical Monte Carlo simulations.** GCMC simulations using the RASPA2.0 code[34]. Equilibration used $5 \times 10^5$ cycles and the subsequent $1 \times 10^6$ cycles were sampled, for each pressure point. Temperature was set to 111 K. The van der Waals interactions for the framework used the UFF force field[78] and methane the united-atom TraPPe force field[79]. Parameters for framework-gas interactions were obtained by Lorentz–Berthelot mixing rules. No charges were considered for the framework atoms. Attributing each methane molecule to each of the pores was achieved via a python script using the pymatgen[80] structure object (Supplementary Fig. 76). Detailed information on the analysis of adsorbate distribution is provided in the supplementary information.

## Data availability

Data used to support the findings of this study are provided in the manuscript or supplementary information. The raw data that support the findings of this study are available from the corresponding author. Single crystal structures can be obtained free of charge from the Cambridge Crystallographic Data Centre via www.ccdc.cam.ac.uk/data_request/cif. CCDC-1889257, 1889255 and 1889256 contain the supplementary crystallographic data for DUT-46, DUT-50 and DUT-151*int*, respectively.

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

## Acknowledgements

This project has received funding from the European Research Council (ERC) under the European Union's Horizon 2020 research and innovation programme (grant agreement No. 742743). The authors thank the BMBF (No. 05K16OD1) and ANR/DFG Programme FUN for financial support and Helmholtz-Zentrum Berlin für Materialien und Energie for allocated beam time at KMC-2 (in situ PXRD), MX BL14.3 (singe crystal X-ray diffraction) beamlines of BESSY II, E9 (FIREPOD) neutron powder diffractometer (in situ NPD) of BER II and travel funding. Furthermore, the MADIREL authors have received funding from the European Union's Horizon 2020 research and innovation programme under the Marie Sklodowska-Curie grant agreement no 641887 (project acronym: DEFNET). G.M. thanks Institut Universitaire de France for its support.

## Author contributions

S. Krause synthesised, activated and performed characterisation of organic ligands and MOF samples. E.T. contributed to the synthesis of organic ligands. S. Krause, V.B., D.W., and D.M.T. contributed to in situ PXRD measurements. S. Krause, V.B., D.W., and A.F. contributed to in situ NPD measurements. V.B. performed refinement of SCD and PXRD data. J.G. performed refinement of NPD data. S. Krause, V.B., I.S. and S. Kaskel contributed to analysis, interpretation and discussion of adsorption and single crystal X-ray diffraction data. F.K. and E.B. performed and discussed NMR data. P.I. and P.L. performed and analysed in situ calorimetry experiments. P.G.Y. and G.M. performed, analysed and interpreted the mercury intrusion experiments. J.D.E. and F.-X.C. performed computational analysis of mechanical and adsorption properties. S.E. performed SEM analysis. S. Krause, V.B., J.D.E. and S. Kaskel organised the project. All authors contributed to writing and improving the manuscript.

## Additional information

**Competing interests:** The authors declare no competing interests.

