## [Peer Review File · Nature Communications]

Reviewers' comments:

Reviewer #1 (Remarks to the Author):

Kaskel and co-workers have produced a well designed, thorough study into NGA materials. This work expands on the previous prototype NGA material DUT-49, providing the community with rationale rules for engineering NGA. The work involves: (1) characterising the mechanical properties of the isorecticular series of related DUT-49 materials by mercury intrusion experiments, computational methods, (2) adsorption studies to understand how guests interact with the frameworks to induce (or not) NGA, via gas adsorption, in-situ calorimetry, in-situ PXRD, NMR, in-situ neutron diffraction and GCMC simulations.

The paper is well written and combines a wealth of analytical techniques to understand a counter-intuitive phenomenon which may be more widespread than previously thought. I would definitely recommend this manuscript for publication after addressing a few small points below.

Pg. 4 Fig 2 and Pg 5 line 141: "The experimental pressures follow the trend of the simulated values but are found to be in general lower".

Upon inspection of Fig 2d, it seems that all have good fit between experimental and simulation, except DUT-48. Is there an explanation for this structure demonstrating via simulation such an elevated pressure for compression? Could this be a structural feature which the simulated model does not capture? Is DUT-48 completely defect free? Is there a way to quantify this? It is widely known that increasing defect concentration can have an effect on softening the mechanical properties of MOFs. In addition, for DUT-48, there is a large discrepancy in the volume change upon compression compared to simulation. It would seem the force-field oversimplifies the volume change, of increasing pore volume, increases ΔV . However, the experimental values show a much less consistent behaviour. Is the force-field model, developed for DUT-49, transferable for this isorecticular series? Or is mercury intrusion experiments a valid way to probe the change in pore volume? Would high-resolution hp x-ray powder diffraction or high-pressure PDF measurements be better methods?

Pg. 7 Line 187-189 & Fig SI 16. You mention that a small hysteresis at 0.06-0.11 p/p₀ indicates a possible displacement of the interpenetrated nets of DUT-151-int. Have you studied this further? I see in pg 8 236- pg 9 Line 249, you carry out in-situ PXRD, and can index a phase shift from C2/c to P-1 structure. Perhaps, it is not interesting for NGA applications, but might show increased selectivity in gas separation for particular guests which bind strongly upon net displacements?

In SI Fig 20, Argon shows a lower pressure transformation to OP phase than n-butane, are there any reasons for this? Perhaps it is just a temperature effect. In addition, there appear to be intermediate phases during the desorption of Ar. Have you been able to characterise these phases further?

Rietveld refinement of CD₄-loaded DUT-48, -49 and -50, supplementary Figs 36-57, high angle peaks at ~74 and ~88 degrees which are large peaks that are not modelled, in any of the structures. Is there a reason for the model to not include these peaks in the refinement? Surely these high-angle peaks would help the refinement in the ordering of the CH₄ molecules within the pores?

Whilst purely cosmetic, in Figure 5, it is impossible to locate the experimental positions of CD₄, perhaps having a more transparent simulated density will aid this, or just side-by-side comparisons of simulated and experimental in the SI would suffice. In addition, the labels should read CD₄ not CH₄. I found the gif of CD₄ filling in the frameworks very helpful, but they also suffer from an opaque simulated density, making comparison between experimental and adsorbed sites difficult. In addition, perhaps I do not fully understand the RDA carried out in the SI. You calculate which pore the CH₄ are in from snapshots and repeated for structures resulting from the in situ CD₄ neutron diffraction experiments. From this data, are you able to compare your

experimental and simulated positions? If I have understood, you use the NPD data as another observation to calculate the radial distribution of methane in each pore. I find this process very elegant, however, from the gif and figure 5, it is hard to see how well the GCMC model captures experimentally determined CD4 positions. I wonder if it can be used to show validation of the force field parameters?

Reviewer #2 (Remarks to the Author):

The manuscript builds upon earlier work on negative gas adsorption (NGA) as reported in ref 3 by the same team and aims to explore the design criteria for new MOFs showing NGA properties. Indeed, another MOF, DUT-50 was identified as the 2nd example of MOF showing NGA. A whole pipeline of characterisation methods have been used, including modelling, DRIFTS, C-NMR, synchrotron X-ray diffraction and in situ neutron powder diffraction. Particularly, computational work has been done in a comprehensive way, supporting/elaborating experimental data as appropriate. Analysis of "detailed structural information" is probably over-killing for systems bearing such high level of structural flexibility, and the corresponding analysis and discussion were conducted all suitably. Exploring structural flexibility in MOFs (including NGA) is a hot topic of research and this work is a nice contribution to this field. I recommend publication of this work following a few revisions.

1. Section: in silico studies. A number of predications were given but they seem to be different to the experimental observations later on (which is fine), but this needs to be rationalised, as least with some plausible explanations. For example, all MOFs show this bi-stable profile, but only 49 and 50 show the actual NGA. Another example "ligand elongation in this system should favour adsorption-induced structural contraction and potentially NGA". Later on, DUT-50 shows higher transition pressure for Ar and methane than that of DUT-49.

2. Synthesis section. the authors claim the interpenetrated structure of -151 will reveal the impact on structural contraction and NGA. This is only critically possible if the iso-structural (cubic) non-interpenetrated -151 can be synthesised and tested.

3. Page 9, "the values of $\Delta\Delta_{adsH_{total}}$ ". I do not quite understand this term. is this the derivative of " Δ_{adsH} ". If so, the numbers (as per unit cell) seem to be very high.

Reviewer #3 (Remarks to the Author):

This is a good article in the subject, but not a huge jump in understanding, as the authors state. The authors already published the concept in nature for example, and have various articles on the microscopic origins already. This article zooms in on some of the details, extends it a bit. In this link there is a paper of the same authors in which they already claimed to explain the microscopic origins:

<https://www.sciencedirect.com/science/article/pii/S2451929416302315>

The effect of interpenetration making the structure more stable to structural changes is well known too.

We would like to thank the referees for their time and valuable comments, which have helped to improve the manuscript. In the following, we have addressed the questions raised and reference the appropriate changes to the manuscript and supplementary information. Changes are marked in red in the revised manuscript and supporting information.

Reviewer 1

Kaskel and co-workers have produced a well designed, thorough study into NGA materials. This work expands on the previous prototype NGA material DUT-49, providing the community with rationale rules for engineering NGA. The work involves: (1) characterising the mechanical properties of the isoreticular series of related DUT-49 materials by mercury intrusion experiments, computational methods, (2) adsorption studies to understand how guests interact with the frameworks to induce (or not) NGA, via gas adsorption, in-situ calorimetry, in-situ PXRD, NMR, in-situ neutron diffraction and GCMC simulations.

The paper is well written and combines a wealth of analytical techniques to understand a counter-intuitive phenomenon which may be more widespread than previously thought. I would definitely recommend this manuscript for publication after addressing a few small points below.

Pg. 4 Fig 2 and Pg 5 line 141: "The experimental pressures follow the trend of the simulated values but are found to be in general lower".

Upon inspection of Fig 2d, it seems that all have good fit between experimental and simulation, except DUT-48. Is there an explanation for this structure demonstrating via simulation such an elevated pressure for compression? Could this be a structural feature which the simulated model does not capture? Is DUT-48 completely defect free? Is there a way to quantify this? It is widely known that increasing defect concentration can have an effect on softening the mechanical properties of MOFs. In addition, for DUT-48, there is a large discrepancy in the volume change upon compression compared to simulation. It would seem the force-field oversimplifies the volume change, of increasing pore volume, increases ΔV . However, the experimental values show a much less consistent behaviour. Is the force-field model, developed for DUT-49, transferable for this isoreticular series? Or is mercury intrusion experiments a valid way to probe the change in pore volume? Would high-resolution hp x-ray powder diffraction or high-pressure PDF measurements be better methods?

Response to reviewer: As the structures of DUT-48 and DUT-50 are very similar to DUT-49, containing no additional chemical functionality other than the aromatic system present in DUT-49, there is no particular reason to believe the forcefield cannot treat these extended (and condensed) analogues with similar accuracy. Moreover, there appears to be a good agreement between the simulated deformation of the ligand in DUT-48 and an experimental single crystal structure of a related compound, PCN-81, see reference: W. Lu et al. [10.1039/C2DT32479B] and S. Krause. et al. [10.1021/acs.jpcc.8b04549].

On the topic of defects, we are currently completing separate work that addresses the influence of defect contraction on the NGA behaviour of DUT-49. We find that only very high concentrations of defects (beyond what we believe potentially occurs during conventional synthesis of these frameworks) produced a considerable effect on NGA and mechanical stability of the framework. We have however not performed any analysis to further inspect potential defects in the presented

systems other than the presented characterisations (TGA, PXRD, gas adsorption, SEM, IR), which indicate no presence of large number of defects.

One possible explanation for this variance in ΔV is the following: The closed pore structures identified by simulation correspond to intermediate or metastable states, whereas the volumes measured by Hg intrusion relate to more compressed structures following bond breakage not captured by these classical simulations. Furthermore, we have demonstrated the differences in ΔV for DUT-48 and DUT-49 originate from amorphization of the samples upon hydrostatic compression with Hg as a pressure transmitting medium [10.1021/acs.jpcc.8b04549]. As the contracted pore (*cp*) phase in all solids investigated exhibits free pore volume, a compression beyond the proposed *cp* phases to even denser, amorphous phases is observed. This seems to be more pronounced for DUT-48 than for the other materials in the series.

The structure and formation mechanism for these dense amorphous phases at this time remains unknown and we are currently attempting, as suggested by the referee, to tackle this by advanced in situ diffraction/compression experiments, which are however out of the scope of this current study. To further apply in situ compression experiments the current challenge is to find a pressure transmitting medium that is not capable of entering the mesopores of the MOF while applying pressures in the range of 1-100 MPa. Media other than Hg have been tested with little success so far. Thus, we believe that the observed deviation is not a consequence of unsuitable MD-simulations but rather different behaviour of the materials in the compression experiments.

Changes to manuscript. Page 6, Line 146: Added *“Previously, we have demonstrated the differences between experimental and simulated compression for DUT-48 and DUT-49 originating from amorphization of the samples. Hg intrusion appears to produce greater compression possibly causing bond breakage not captured by the classical simulations used.”*

Pg. 7 Line 187-189 & Fig SI 16. You mention that a small hysteresis at 0.06-0.11 p/p₀ indicates a possible displacement of the interpenetrated nets of DUT-151-int. Have you studied this further? I see in pg 8 236- pg 9 Line 249, you carry out in-situ PXRD, and can index a phase shift from C2/c to P-1 structure. Perhaps, it is not interesting for NGA applications, but might show increased selectivity in gas separation for particular guests which bind strongly upon net displacements?

Response to reviewer: As pointed out by the reviewer, the flexibility of DUT-151, although different to the rest of the series, is very interesting. We have used the in situ PXRD data mentioned by the referee to refine the crystal structures via Rietveld refinement. The different structures are displayed in Supplementary Figure 31 (page 50) and cif files were submitted to the CCDC database under the codes 1890381, 1890382, and 1890383, which contain the structural data for DUT-151int-act, DUT-151int-cp and DUT-151int-op, respectively. These structural data are also summarized in the Supplementary Information (page 48). We agree with the referee that this behaviour can indeed be interesting for separation and storage of small gas molecules. Thank you for this valuable suggestion! We will explore this further, however, it remains beyond the scope of the presented investigation.

In SI Fig 20, Argon shows a lower pressure transformation to OP phase than n-butane, are there any reasons for this? Perhaps it is just a temperature effect. In addition, there appear to be intermediate phases during the desorption of Ar. Have you been able to characterise these phases further?

Response to reviewer: The difference in pressure, at which the transition occurs, is a result of the different adsorption temperature. Argon experiments were carried out at 87 K (at which p₀ is 100 kPa), while n-butane experiments were carried out at 303 K (at which p₀ is 230 kPa).

We are currently further exploring the influence of adsorption temperature, which has been observed in our previous work on NGA in DUT-49 [10.1038/nature17430]. Considering that the *op* →

cp transition occurs for several gases in the relative pressure range of 0.1-0.15 the absolute pressure of this transition strongly depends on temperature.

In fact, the stepwise isotherm in the desorption branch of DUT-50 suggests the formation of intermediate phases, as pointed out by the referee. We have previously identified these intermediate phases upon desorption of methane at 111 K in DUT-49 [10.1038/nature17430] characterized by in situ PXRD and refined by Rietveld refinement but not yet in the case of Argon desorption.

Changes to manuscript. Page 9, Line 237: Added “Moreover, during desorption there is evidence for intermediate pore phases, similar to those observed in DUT-49.”

Rietveld refinement of CD₄-loaded DUT-48, -49 and -50, supplementary Figs 36-57, high angle peaks at ~74 and ~88 degrees which are large peaks that are not modelled, in any of the structures. Is there a reason for the model to not include these peaks in the refinement? Surely these high-angle peaks would help the refinement in the ordering of the CH₄ molecules within the pores?

Response to reviewer: As indicated in the Supporting Information (page 52), the peaks at 75 ° and 89 ° originate from the Aluminium sample holder. Due to experimental restrictions these peaks cannot be omitted. Rather, these peaks were used as an internal standard to normalize the patterns and monitor the increase of intensity of the main reflections. As these peaks do not originate from the MOF samples, they were omitted from the Rietveld refinement.

Whilst purely cosmetic, in Figure 5, it is impossible to locate the experimental positions of CD₄, perhaps having a more transparent simulated density will aid this, or just side-by-side comparisons of simulated and experimental in the SI would suffice. In addition, the labels should read CD₄ not CH₄. I found the gif of CD₄ filling in the frameworks very helpful, but they also suffer from an opaque simulated density, making comparison between experimental and adsorbed sites difficult.

Response to reviewer: We have attempted to make this comparison using a more transparent representation of the density, however, this did not make for a clear comparison given the three-dimensional distribution of density and adsorbates. For a clear comparison we recommend to refer the RDA comparisons added as Supplementary Figures 79-81.

In addition, perhaps I do not fully understand the RDA carried out in the SI. You calculate which pore the CH₄ are in from snapshots and repeated for structures resulting from the in situ CD₄ neutron diffraction experiments. From this data, are you able to compare your experimental and simulated positions? If I have understood, you use the NPD data as another observation to calculate the radial distribution of methane in each pore. I find this process very elegant, however, from the gif and figure 5, it is hard to see how well the GCMC model captures experimentally determined CD₄ positions. I wonder if it can be used to show validation of the force field parameters?

Response to reviewer: You are correct that it is possible to compare the experimental CD₄ and simulated CH₄ positions using the RDA approach described. In practise, however, the comparison does not provide a validation. The refinement procedure produces an occupancy weighted singular structure. This static picture of adsorption is difficult to compare to the distribution of positions from simulations. This led us to compare only the populations of each pore.

Nevertheless, we have included an RDA comparison and importantly there is overlap observed between experimental positions and the simulations distributions.

Changes to Supporting Information. Added Supplementary Figures 79-81.

Reviewer 2

The manuscript builds upon earlier work on negative gas adsorption (NGA) as reported in ref 3 by the same team and aims to explore the design criteria for new MOFs showing NGA properties. Indeed, another MOF, DUT-50 was identified as the 2nd example of MOF showing NGA. A whole pipeline of characterisation methods have been used, including modelling, DRIFTS, C-NMR, synchrotron X-ray diffraction and in situ neutron powder diffraction. Particularly, computational work has been done in a comprehensive way, supporting/elaborating experimental data as appropriate. Analysis of "detailed structural information" is probably over-killing for systems bearing such high level of structural flexibility, and the corresponding analysis and discussion were conducted all suitably. Exploring structural flexibility in MOFs (including NGA) is a hot topic of research and this work is a nice contribution to this field. I recommend publication of this work following a few revisions.

1. Section: in silico studies. A number of predications were given but they seem to be different to the experimental observations later on (which is fine), but this needs to be rationalised, as least with some plausible explanations. For example, all MOFs show this bi-stable profile, but only 49 and 50 show the actual NGA. Another example "ligand elongation in this system should favour adsorption-induced structural contraction and potentially NGA". Later on, DUT-50 shows higher transition pressure for Ar and methane than that of DUT-49.

Response to reviewer: We thank the referee for pointing this out. Although all solids investigated (except DUT-151int) exhibit bistability, only DUT-49 and DUT-50 demonstrate phase changes during adsorption (and indeed NGA). The adsorption energy gained by contraction of the pores for DUT-48 and DUT-46 is not sufficient to compensate the greater energy required for this transition. This greater energy (or stress) to contract DUT-48 and DUT-46 was demonstrated by simulation and hydrostatic compression experiments, using Hg as pressure transmitting medium. This is described in the manuscript (Page 12, Line 348).

In contrast to hydrostatic compression, adsorption induced contraction strongly depends on the adsorption mechanism which we in detail investigate in this study. We suggest that the transition pressure of the $op \rightarrow cp$ transition directly correlates to the filling of the mesoporous cavities. With increasing pore size, the pressure at which the pore filling occurs is shifted to higher pressures. This is also well known from other mesoporous solids [10.1016/S1387-1811(03)00339-1]. Thus, for DUT-50 the adsorption-induced transition is expected to occur at higher gas pressures while the compressive hydrostatic pressure applied in the intrusion experiments is lower compared to DUT-49 due to the decreasing mechanical stability of DUT-50. The manuscript was amended to provide an explanation for this observation.

Changes to manuscript. Page 13, Line 360: Added "This is consistent with DUT-50 exhibiting a higher p_{NGA} than DUT-49, owing to the large pore sizes responsible for delayed mesopore filling."

2. Synthesis section. the authors claim the interpenetrated structure of -151 will reveal the impact on structural contraction and NGA. This is only critically possible if the iso-structural (cubic) non-interpenetrated -151 can be synthesised and tested.

Response to reviewer: We were unable so far to obtain a non-interpenetrated sample of DUT-151, despite our best efforts. However, the detailed simulations presented demonstrate a lack of a cp phase, following the application of hydrostatic pressure. Given the investigated materials are isorecticular we believe this provides the evidence required to make these claims. Nevertheless, the

claims regarding interpenetration have been modified in the Supporting Information to better reflect our findings.

Changes to Supporting Information. Page 51: Text modified “DUT-151/*int* lacks the metastable character well reflected by simulations of the free energy profile and absence of ligand buckling.” and “In conclusion interpenetration is suggested to suppress contraction and NGA for this topology, however, this remains untested experimentally for a non-interpenetrated DUT-151 material.”

3. Page 9, "the values of $\Delta\Delta_{ads}H_{total}$ ". I do not quite understand this term. is this the derivative of " $\Delta_{ads}H$ ". If so, the numbers (as per unit cell) seem to be very high.

Response to reviewer: We apologise that the term $\Delta\Delta_{ads}H_{total}$ was not clearly introduced in the main text. The value of $\Delta\Delta_{ads}H$, as introduced in our previous work on NGA [10.1038/nature17430], describes the difference in enthalpy of adsorption between the *op* and *cp* phases, per unit cell. This value does not represent a derivative of $\Delta_{ads}H$ and was defined in the Supplementary Information (page 92), however, we have updated the manuscript to more clearly define this term in the main text.

Changes in manuscript. Page 9, Line 184: Sentence modified to “The difference in adsorption enthalpy between the *op* and *cp* structures ($25.4\pm 2 \text{ kJ mol}^{-1}$) is multiplied by the number of *n*-butane molecules (n_{ads}) adsorbed per unit cell in the *op* phase to give $\Delta\Delta_{ads}H_{total}$, which approximates the gain in enthalpy per unit cell upon structural contraction.”

Reviewer 3

This is a good article in the subject, but not a huge jump in understanding, as the authors state. The authors already published the concept in nature for example, and have various articles on the microscopic origins already. This article zooms in on some of the details, extends it a bit.

In this link there is a paper of the same authors in which they already claimed to explain the microscopic origins:

<https://www.sciencedirect.com/science/article/pii/S2451929416302315>

The effect of interpenetration making the structure more stable to structural changes is well known too.

Response to reviewer: We appreciate the comment of the referee. In fact, our previous articles on the subject mentioned demonstrate the realisation of NGA, and subsequently a progressive understanding of this phenomenon. Nevertheless, NGA remains a very complex process that up to this point was only discussed for a single framework, DUT-49. The present article describes the culmination of these previously published techniques (and understanding) to characterise NGA for a series of isorecticular materials, allowing the exploration of design criteria for new NGA materials. From this careful investigation, we succeed in demonstrating NGA in DUT-50, which is only the second material to exhibit this phenomenon, making this material alone interesting and unique. Furthermore, we demonstrate a novel technique to analyse the pore filling mechanism in a hierarchical crystalline pore structure, allowing us to connect the moment of NGA to the filling of mesoporous cavities. We believe this advance will outline other potential NGA materials and also link metastable phenomena, such as capillary condensation, to NGA. In addition, we feel that the applied radial distribution analysis can be a valuable tool for investigations in other fields such as gas storage and separation.

We agree that interpenetration is known to increase the mechanical stability of the framework mainly due to a reduction in free pore volume, however, there are few investigations, to our knowledge, that systematically link interpenetration to adsorption-induced transitions (or other related phenomena) in MOFs or other porous solids. Additionally, DUT-151int exhibits a pore volume in the range of DUT-48 and could by this means, in principle, demonstrate NGA. Using a combination of simulation and in situ experiments, we clearly demonstrate the factors responsible for the absence of NGA in DUT-151int.

REVIEWERS' COMMENTS:

Reviewer #1 (Remarks to the Author):

Thank you to the authors for detailed discussion and explanation of all of the previous comments. I think this work well done, and I recommend it for publication.

Reviewer #2 (Remarks to the Author):

The authors have addressed my concerns and I am happy to recommend for publication of this work.